# Assessing Heat–Health Vulnerability Through Temporal, Demographic, and Spatial Lenses: A Time-Stratified Case-Crossover Analysis in New York State

**DOI:** 10.3390/ijerph22071124

**Published:** 2025-07-16

**Authors:** Heather Aydin-Ghormoz, Temilayo Adeyeye, Wanhsiang Hsu, Neil Muscatiello

**Affiliations:** Center for Environmental Health, New York State Department of Health, Albany, NY 12237, USAwanhsiang.hsu@health.ny.gov (W.H.);

**Keywords:** climate, heat-related illness, case-crossover

## Abstract

New York State (NYS) has experienced warming outpacing the national average, and heat events are increasing. This case-crossover study uses conditional logistic regression to estimate how maximum heat index affects heat-related illness across temporal, demographic, and spatial groups in NYS, from May to September, 2008–2019. The highest risks were in May (Relative Risk (RR) = 1.81, CI: 1.72, 1.90) and August (RR = 1.86, CI: 1.79, 1.94). Older adults, especially those aged 85 and above, are at greatest risk (RR = 1.83, CI: 1.71, 1.96). The Southern Tier climate region had a higher risk (RR = 1.93, CI: 1.80, 2.07) than several other regions. Overall, similar risk between rural and urban NYS was observed. Rural non-Hispanic Black (RR = 2.38, CI: 1.78, 3.19) populations had a higher risk than their urban counterparts. This study was innovative for NYS, providing a deeper understanding of heat–health risks to vulnerable sub-groups. This can assist with facilitating targeted interventions and public health messaging during the periods of highest risk, such as promoting awareness of cooling centers and air-conditioning benefits.

## 1. Introduction

Annual average and maximum temperatures have been rising in New York State (NYS) since the early 20th century [1,2]. Climate projections indicate that both overall temperature and the frequency of heat events will continue to increase throughout the 21st century [1,2]. NYS is warming at a rate that has outpaced the national average, with heat events projected to increase in frequency statewide [2]. While New York City (NYC) is projected to remain the warmest region, temperature increases and extreme heat events are also anticipated in historically cooler areas [2].

As temperatures rise and heat events become more frequent, the risks of heat-related illnesses (HRIs) for residents of NYS will also increase. Numerous studies indicate elevated risk of HRIs in specific demographic groups [3,4,5,6,7]. An individual’s racial and ethnic identity is often correlated with heat-related morbidity and mortality in the United States [3]. However, this association has been found to be strongly driven by social determinants of health [8,9]. The inequitable distribution of heat–health burden across various geographic locations, demographic characteristics, and occupations is further exacerbated by disparities in access to protective factors both at the individual and community level [10]. Increased age is often associated with heat-related morbidity and mortality, especially among the elderly. Older people may have age-related vulnerabilities such as decreased thermoregulatory function and comorbidities [6,7]. The working age population has also been found to be impacted by HRIs, which may be associated with increased time spent outdoors for work and/or recreation [4,11]. Additionally, males are often identified as having an increased association with HRIs [11,12].

A previous study of associations between hospitalizations/emergency department (ED) visits and temperature metrics in NYS found that rural regions of NYS had a similar risk to urban regions. This analysis prompted the National Weather Service (NWS) to lower the advisory threshold for the upstate region of NYS and surrounding areas [13]. NYS is one of the nation’s most populous states, with most of the population living in urban areas; however, its geography is largely rural. We included an assessment of heat risk in the varying climate regions, since these areas vary in demographic diversity, population concentration, and the prevalence of outdoor activities. Historically, in the literature, heat–health effects have been portrayed as a predominantly urban issue; however, recent research shows that rural communities are also vulnerable. Urban and rural risk was assessed to confirm previous findings showing similar risk in NYS [13]. This study builds upon previous work [14] by analyzing additional years of data to better understand the differential risk of HRIs within select temporal, demographic, and spatial groups

## 2. Materials and Methods

### 2.1. Study Design and Population

Using administrative claims data, a case-crossover study design was used to assess overall risk for HRI, stratified by several temporal, demographic, and spatial factors. Specifically, month, time period, age, sex, race/ethnicity, urbanicity, climate region, and counties with major cities. The study population includes patients seen at the ED or hospitalized for HRIs in NYS during May–September of 2008 through 2019.

### 2.2. Health Outcomes

The main heat–health outcome evaluated was HRIs (ICD-9: 992, E900.0, E900.9; ICD-10: T67, X30, X32). Encounter-level health data for daily May–September ED visits and hospitalizations were acquired from the NYS Statewide Planning and Research Cooperative System (SPARCS). SPARCS is a statewide all-payer database of patient information, including primary and secondary diagnoses for inpatient and outpatient (ambulatory surgery, ED, and outpatient services), excluding Federal and Veterans Health Administration hospitals, Indian Health Service hospitals, and institutionalized populations [14]. We restricted our analysis to unscheduled visits, specifically ED visits and hospitalizations. A scheduled outpatient visit, or ambulatory surgery, is more likely to take place several days after the onset of a health outcome, and using this date to assign temperature exposure may result in exposure misclassification. Reoccurrence of ED visit and/or hospitalization within a one-week period following a previous encounter was excluded, and any reoccurrence more than one week following a previous encounter was considered a new event.

### 2.3. Air Temperature

Maximum heat indices (HI_max_) (in degrees Fahrenheit) from 2008 to 2019 were derived from the National Aeronautics and Space Administration’s (NASA) North American Land Data Assimilation System forcing data for Phase 2 (NLDAS-2) [15]. NLDAS-2 provides quality-controlled, spatially and temporally consistent North American land-surface model data from observation and reanalysis in ~12–14 km (1/8th degree) grid spacing with an hourly temporal resolution [15]. The non-precipitation land surface forcing fields are derived from the analysis fields of the National Centers for Environmental Prediction (NCEP) North American Regional Reanalysis (NARR) [15]. NARR analysis fields have a 32 km spatial resolution and 3 h temporal resolution and are spatially interpolated to the 1/8th degree and temporally disaggregated to the NLDAS-2 hourly resolution [15,16]. NLDAS-2 forcing data used in this study can be downloaded from the Goddard Earth Sciences (GES) Data and Information Services Center (DISC) [17]. We derived population-weighted HI_max_ at the 2010 Census tract level by first determining the grid cell of each block group centroid, assigning a weight based on the amount of population living in the block group, then calculating a population-weighted HI_max_ across NYS Census tracts. Exposure on the day of the ED visit or hospitalization for HRIs was assigned using the patient’s residential address. Geocoded residential addresses were linked by 2010 Census tract and date to daily HI_max_. Geocoding of residential addresses was carried out using ArcGIS Desktop (ArcMAP), version 10.8.2 (Environmental Systems Research Institute (ESRI), Redlands CA, USA).

### 2.4. Air Pollution

Our models adjusted for fine particulate matter (PM_2.5_) and ozone using data from the Environmental Protection Agency (EPA) downscaler (DS) model [18]. The DS model fuses monitor data from the National Air Monitoring Stations/State and Local Air Monitoring Stations with 12 km gridded output from the Models-3/Community Multiscale Air Quality Model (CMAQ) [18]. Patient exposure to PM_2.5_ and ozone was assigned using the patient’s geocoded residential address and linked by 2010 Census tract and date.

### 2.5. Stratification Factors

Individual patient demographic variables were provided by SPARCS, including patient age, sex, race, and ethnicity. Spatial factors based on patients’ residential address include climate region and urbanicity. Climate regions were defined based on the regions in the NYS Climate Impacts Assessment [2]. Counties that fell into two climate regions were assigned to a region using population weighting. The final regions were Adirondacks: Lewis, Hamilton; Catskills: Delaware, Greene, Sullivan, Ulster; Central/Finger Lakes: Wyoming, Livingston, Ontario, Yates, Seneca, Schuyler, Tompkins, Cortland, Onondaga; Champlain Valley: Clinton, Essex, Warren; Great Lakes: Jefferson, Oswego, Cayuga, Wayne, Monroe, Orleans, Genesee, Niagara, Erie, Chautauqua; Long Island: Nassau, Suffolk; Mohawk River Valley: Madison, Oneida, Herkimer, Fulton, Montgomery, Otsego, Schoharie; New York City: Bronx, Queens, Richmond, New York, Kings; North Hudson: Columbia, Albany, Schenectady, Saratoga, Washington, Rensselaer; South Hudson: Orange, Putnam, Rockland, Westchester, Dutchess; Southern Tier: Cattaraugus, Allegany, Steuben, Chemung, Tioga, Broome, Chenango; and St Lawrence Valley: St Lawrence, Franklin (Figure 1). Urbanicity was defined using rural–urban commuting area (RUCA) codes, which classify census tracts from metropolitan to rural on a ten-point scale using measures of population density, urbanization, and journey to work commuting [19]. We used categorization C: Urban 1.0, 1.1, 2.0, 2.1, 3.0, 4.1, 5.1, 7.1, 8.1, and 10.1; Rural 4.0, 4.2, 5.0, 5.2, 6.0, 6.1, 7.0, 7.2, 7.3, 7.4, 8.0, 8.2, 8.3, 8.4, 9.0, 9.1, 9.2, 10.0, 10.2, 10.3, 10.4, 10.5, and 10.6 [20]. The most recent RUCA codes are based on the 2010 decennial census and the 2010 ACS (American Community Survey) 5-year estimates [19]. Additionally, cases were assessed temporally by warmer months in NYS, specifically May through September.

### 2.6. Statistical Analysis

A case-crossover analysis with a semi-symmetric bidirectional, time-stratified design was used to assess the association of HI_max_ on ED visits and hospitalizations in NYS from May to September. This study design compares HI_max_ on the day of ED visit or hospitalization (case/exposure day) with HI_max_ on days before or after (control period), within a pre-specified stratum window of time when the case patient is not in the ED or hospitalized [13,21]. The control period is ±7, ±14, or ±21 days from the exposure day within the same-month stratum window, which provides each case up to four control days. This control period selection identifies control days restricted to the same day of the week, month, and year to control for seasonal and day of week variation [13,22].

A case-crossover study design is ideal for acute onset health outcomes related to short-term exposures and is therefore well suited for estimating heat–health risk. Additionally, a case-crossover study design is efficient, statistically powerful, and controls for non-time-varying confounding factors, since each case serves as its own control [23]. A limitation of the study design may arise from time-varying confounding. We utilized time stratification and controlled for time varying PM_2.5_ and ozone in the models to address this, consistent with previous methods [13]. Air pollution is unlikely to be associated with HRIs independently; however, the literature supports it may have interactive effects on health outcomes when combined with extreme heat by driving oxidative stress and inflammation in the body [24,25,26,27].

The annual rates of HRIs per 100,000 population were calculated using 2010 ACS 5-year estimates. Conditional logistic regression models were stratified by demographic, spatial, and temporal factors to determine and compare the relative risk of HRIs for a 5 °F change in HI_max_. We also carried out group comparisons of risk ratios to evaluate the associations of predictor variables with the outcome across different groups. All statistical analyses were performed using SAS^TM^ software Version 9.4 (SAS Institute Inc., Cary, NC, USA).

## 3. Results

There were 23,853 HRI ED visits (86%) or hospitalizations (14%) in NYS from May to September, 2008–2019. Most cases (45%) of HRIs occurred in July. Rates of HRIs tended to increase with age, with the highest annual rate of HRIs occurring in individuals 85 years and older (22.2 per 100,000 population). Cases were most common in males (59%) and White, non-Hispanic (57%) individuals (Table 1). Males also had greater rates of HRIs (12.2), being about 40% higher than the rate for females (8.1). The rate for Black, non-Hispanics (11.8) was between 15 and 55% higher than White, non-Hispanics (10.2), and Hispanics (6.7). Reflecting the distribution of the population in NYS, a higher proportion of HRI cases were found in urban NYS (87%) compared with rural NYS (13%), although rates of HRIs were slightly higher in rural areas (11.3 in rural and 10.0 in urban areas, respectively). The lowest rate of HRI was in NYC (7.5), while the highest rates were observed in the St. Lawrence Valley (16.5), Southern Tier (15.2), and Champlain Valley (15.1). Based on the high population counts in the rate calculations, rates with narrow confidence intervals are statistically significantly different within groups. (Table 1).

Table 2 displays associations between HI_max_ and HRIs modified by various factors such as month of exposure, age, sex, and race/ethnicity. In the analysis by month, warmer temperatures within all summer months were statistically significantly associated with HRI, with the largest increased risks observed in the months of May (RR = 1.81, CI: 1.72, 1.90) and August (RR = 1.86, CI: 1.79, 1.94) relative to July (RR = 1.64, CI: 1.60, 1.67), despite the latter being the warmest month in NYS. Similarly, warmer temperatures were statistically significantly associated with HRIs within each age group, with an increasing trend in risk with increasing age. Similar risks were observed across race/ethnicity and sex categories. In the analysis by geographic region, the Southern Tier climate region (RR = 1.93, CI: 1.80, 2.07) of NYS had a statistically significant higher risk than several other regions, specifically North Hudson, NYC, South Hudson, and Great Lakes. The Adirondack climate region (RR = 3.28, CI: 1.68, 6.42) also had a higher risk but was not statistically different than other regions due to wide confidence intervals associated with relatively small counts.

Further breakdown of the demographic differences by urbanicity (Figure 2, Figure 3 and Figure 4) indicated a statistically significant higher risk for rural non-Hispanic Blacks (RR = 2.38, CI: 1.78, 3.19) compared with urban non-Hispanic Blacks (RR = 1.71, CI: 1.65, 1.77) (Figure 3).

Appendix A displays additional sub-analyses of the association between HI_max_ and HRIs among age groups by month. While there were some differences within particular age groups across months, there were no clear patterns. In a sub-analysis of age by month, HRIs risk in 45–64-year-olds was highest in the early and later summer months (Appendix A). In May (RR = 2.17, CI: 1.91, 2.45), risk was statistically significantly greater than in June (RR = 1.76, CI: 1.64, 1.88) and July (RR = 1.65, CI: 1.58, 1.71) (Appendix A).

Additionally, in August, 45–64-year-olds (RR = 1.89, CI: 1.75, 2.03) had increased risk compared with the risk in July (RR = 1.65, CI: 1.58, 1.71) (Appendix A). In September, HRI risk in 45–64-year-olds (RR = 1.95, CI: 1.75, 2.18) was statistically significantly greater than risk in July (Appendix A). Among individuals aged 20–44, risk was highest in August (RR = 1.86, CI: 1.74, 1.98) and statistically significantly higher than the risks in June (RR = 1.76, CI: 1.64, 1.73), July (RR = 1.62, CI: 1.57, 1.68), and September (RR = 1.60, CI: 1.48, 1.73) (Appendix A). In the other age categories, there were no other statistically significant month-to-month comparisons. As shown in Appendix A, there were no clear patterns of the association between HI_max_ and HRIs among demographic groups within other demographic groups.

## 4. Discussion

This study assessed heat–health rates and risks in various demographic groups by month of exposure and geographic locations. This analysis was innovative for NYS, providing a deeper understanding of heat–health risks to vulnerable sub-groups within the state. These findings may assist with facilitating targeted interventions and public health messaging. As expected, increasing temperatures were broadly associated with increased risk of HRIs among all demographic groups, geographic groupings, and summer months. However, the analysis was designed to specifically assess differential risk in subcategories throughout the state.

While HRI rates were highest in July, the increase in the risk of ED visit or hospitalization per unit increase in the heat index was highest in May and August. In NYS, July generally has the highest average heat index, so higher rates of HRIs are expected. Elevated risk in the transition month of May, when the heat index in NYS is beginning to rise, may be associated with early-season deficits in thermoregulation. Thermoregulation refers to the body’s ability to maintain internal temperature homeostasis despite fluctuations in external conditions [28]. As the warm season progresses, individuals are more likely to become increasingly heat-adapted. Heat acclimatization (naturally induced, e.g., physiological seasonal heat adaptation) or heat acclimation (artificially induced, e.g., air-conditioning) occurs as a response to elevated skin and internal temperatures [29]. People could perceive May as being too early in the heat season for heat acclimation, such as the use of air-conditioning [30]. Heat acclimatization occurs from repeated exposure to hot temperatures, leading to effects such as increased sweating efficiency and circulatory stabilization [31]. Supporting that theory are results from this study showing the highest May risks occurred among those aged 45–64, which could be construed as an older worker population. Awareness that there is an increased risk of HRIs in May, when it would be less expected, is useful for public health adaptations and messaging during the early heat season.

Older age is a known risk factor for heat-related illness due to various factors. In this study, HRI rates and risks generally followed an increasing trend from the youngest to the oldest. Older adults have decreased thermoregulatory capacity in response to rapid changes in temperature [32,33]. Additionally, this population has a higher prevalence of comorbidities that increase vulnerability to heat, including cardiovascular, pulmonary, or renal disease, and metabolic conditions such as obesity and Type II diabetes [32,34]. Relatedly, older adults are more likely to be taking medications for comorbidities that may further impair the body’s capacity to thermoregulate, such as diuretics, sedatives, tranquilizers, and some heart and blood pressure drugs [32,34]. In addition to comorbidities, older people may experience diminished access to healthcare services, transportation, and assistance [34].

Previous research in NYS estimated that urban and rural risks for HRIs were similar, despite illness from extreme heat often being portrayed as an urban problem [13]. This analysis confirms and expands on these findings, exploring variation in risk across different demographic groups. This research found an increased risk of HRIs among rural non-Hispanic Black populations. Heat vulnerability is related to exposure to environmental conditions (i.e., heat), sensitivity (e.g., associated with preexisting conditions), and ability to adapt to hazards [35]. However, drivers of heat vulnerability in rural settings may differ from those in urban settings [36]. Urban residents of NYS (outside of NYC) may be more vulnerable to heat due to more immigrant and limited English-proficiency populations, where language may create barriers to resource access and alert messaging [5]. Additionally, urban NYS residents may be more vulnerable to heat exposure due to environmental/urbanicity factors, such as older homes, building intensity, less open land, and housing density, which may all contribute to the urban heat island effect [5].

Conversely, in rural areas, demographic and geographic factors, outdoor occupations employing a higher proportion of the workforce, and accessibility to cooling may drive heat vulnerability [36,37]. The Southern Tier climate region of NYS had both one of the highest rates of HRIs and the highest risk of HRIs compared with several other regions. Potential reasons for these observations may include more heat exposure among outdoor workers, e.g., agricultural, forestry, and fishing industry, in the region [38]. The 2018 Behavioral Risk Factor Surveillance System (BRFSS) estimates the household prevalence of air conditioning, a protective factor for heat-related illness, to be higher in NYC compared with NYS outside of NYC (unpublished data). The economic status of individuals and communities can impact how they respond and react to extreme heat [5,39]. Data from the BRFSS also suggest a trend between income and access to air-conditioning, with a higher proportion of respondents from households making less than USD 50,000 reporting no access to air conditioning, and that even among people with access to air conditioning, the costs of electricity bills prevent some people from using it.

While it is supported in the literature that urban–rural disparities exist in access to healthcare and that racial/ethnic minorities have poorer access to healthcare, there is little published research examining the health outcomes of rural minorities compared with their urban counterparts in NYS. Some research does indicate that the rural non-Hispanic Black population may have greater comorbidity mortality rates and less access to healthcare, screening, and insurance when compared with their urban counterparts [40,41,42]. As previously discussed, increased comorbidities and diminished access to health resources can increase vulnerability to HRI. Notably, estimates from the 2018 BRFSS suggest that rural non-Hispanic Blacks may have lower access to air conditioning than other groups. However, it is important to note that the total NYS population counts of rural non-Hispanic Black individuals in the 2018 BRFSS data and this analysis are relatively small, which may affect these results due to less accuracy. Further research in these populations may be useful to better understand their vulnerability. Rural communities often experience a lack of data that hinders decision-making and the allocation of vital resources [37]. Our findings can contribute to informing evidence-based polices and adaptations within NYS to address the heat-vulnerable populations of rural communities during the heat season.

NYS is engaged in several ongoing adaptation strategies. Concerted efforts to mitigate language barriers to public health resource access and messaging are made by both the NYS Department of Health and the NYC Department of Health and Mental Hygiene, where informational webpages can be translated into several languages common in NYS, but there may be barriers to accessing that information [43,44]. Staff from the NYS Department of Health have been working with the NYS Office of Temporary and Disability Assistance to promote the Home Energy Assistance Program Cooling Benefit, which provides a free window air-conditioning unit to those who meet program criteria [45]. Additionally, eligible members of the New York State of Health’s Essential Plan can avail of the Essential Plan Cooling Program, which also provides a free air-conditioning unit [46]. The NYS Department of Labor has developed guidance aimed at protecting outdoor workers from the effects of extreme heat [47]. NYS recently unveiled a comprehensive extreme heat action plan (EHAP) for statewide action, focusing on equitably addressing extreme heat and its effects, reducing vulnerability, and bolstering community capacity [48]. This plan emphasizes extreme heat adaptation and is distinct from emergency management plans. The roadmap outlines forty-nine actions divided into four goal-oriented tracks: supporting extreme heat adaptation planning and implementation; enhancing preparedness, communication, and workers’ safety; improving the resilience of built environments, infrastructure, and managed spaces; and advancing ecosystem-based adaptation [48]. As the climate changes and extreme heat becomes a threat in previously unaffected areas, climate adaptation is increasingly critical. Building adaptive capacity and enhancing community resilience are essential to mitigate climate change impacts and prepare for future extreme weather events, thereby promoting the well-being of affected communities. With the EHAP in place, New Yorkers will be better prepared and adapted for extreme heat events. These results could be useful in considering recommendations made in the EHAP. Public health messaging and interventions such as the HeatRisk tool can be targeted to high-risk populations. HeatRisk was developed by the NWS in partnership with the CDC and recently expanded to include NYS and could be a useful early-warning resource [49].

The strengths of this study include the ability to use many years of available ED and hospitalization data, allowing for a sample size suitable to assess differential heat–health risks among various sub-categories of NYS residents. This research also has the advantage of using spatially resolved NLDAS heat index data. In NYS, meteorological monitoring sites have historically been sparsely situated and primarily located in urban centers with higher populations. While the NYS Mesonet, a network of over 100 monitors collecting a rich array of meteorological data points, holds promise for the future, that network was completed in 2018, much later than the early years of data used in this study [50]. NLDAS reanalysis data provide greater spatial coverage for estimating individual temperature exposure [13]. Additionally, the case-crossover methodology used in this study controls for confounding by fixed characteristics and same day of week, month, and year control selection controls for seasonal variation.

However, there are some limitations to consider. This research uses a large administrative claims data set to assign outcomes; less severe cases of HRIs who did not seek emergency care are not represented in the analysis. Additionally, there is potential for cases of HRIs to be missing from the data if HRIs were not documented and other diagnostic codes took precedence in the patient record. Even with the range of years in the study, some subgroup analyses yielded wide confidence intervals due to small sample sizes. Additionally, relevant information about the exposure setting (i.e., at home, work, etc.) was unavailable. Exposure to heat index, PM_2.5_, and ozone was assigned using residential address, in the absence of individual information on activity and location on the day of exposure, which may introduce misclassification of the exposure if the exposure were to have happened at a location other than outdoors at an individual’s residence. Relatedly, we do not have information on an individual’s acclimatization or acclimation patterns, such as whether they had access to air conditioning or have greater heat tolerance due to outdoor work. There has been some recent research discussing, in the context of extreme heat exposure, how case-crossover analysis methods may have poorer efficiency than time series analytic methods. However, this analysis spans from 2008 to 2019, providing sufficient case counts.

Addressing the impact of extreme temperatures on health requires a multifaceted approach, including targeted public health interventions and messaging, urban planning and policies aimed at reducing greenhouse gases, and improving access to relevant cooling benefits. As climate change continues to intensify, it is imperative that proactive strategies are adopted to safeguard public health and ensure that no one is left behind.

## 5. Conclusions

In summary, the risk of HRIs increases for individuals as the heat index increases; however, it is important for targeted public health interventions to be aware that some transitional and warmer months, age categories, and geographic locations have increased risk. This research identified greater HRI risk associated with increasing heat index in May and August, among older age groups, and among individuals in the Southern Tier of NYS. Additionally, while urban and rural risk is similar, the study also found greater HRI risk associated with increasing heat index among rural non-Hispanic Blacks compared with their urban counterparts. Ultimately, during periods of increased temperature and heat index, everyone must take care to protect themselves from heat-related illness, especially individuals who belong to greater risk groups.

Within NYS, this analysis was innovative by providing a more nuanced understanding of heat-related health inequities. Understanding the interactions between environmental and socio-demographic determinants of health that may be driving them can help support heat risk reduction and adaptation strategies, future policy decisions, community engagement efforts, and development of innovative solutions in NYS and in states with a similar climate profile. Future research efforts are necessary to assess heat–health impacts by additional socio-demographic factors, potentially focusing on rural non-Hispanic Blacks and outdoor workers, as well as neighborhood disadvantage and its effect on heat-related morbidity and mortality.

## Figures and Tables

**Figure 1 ijerph-22-01124-f001:**
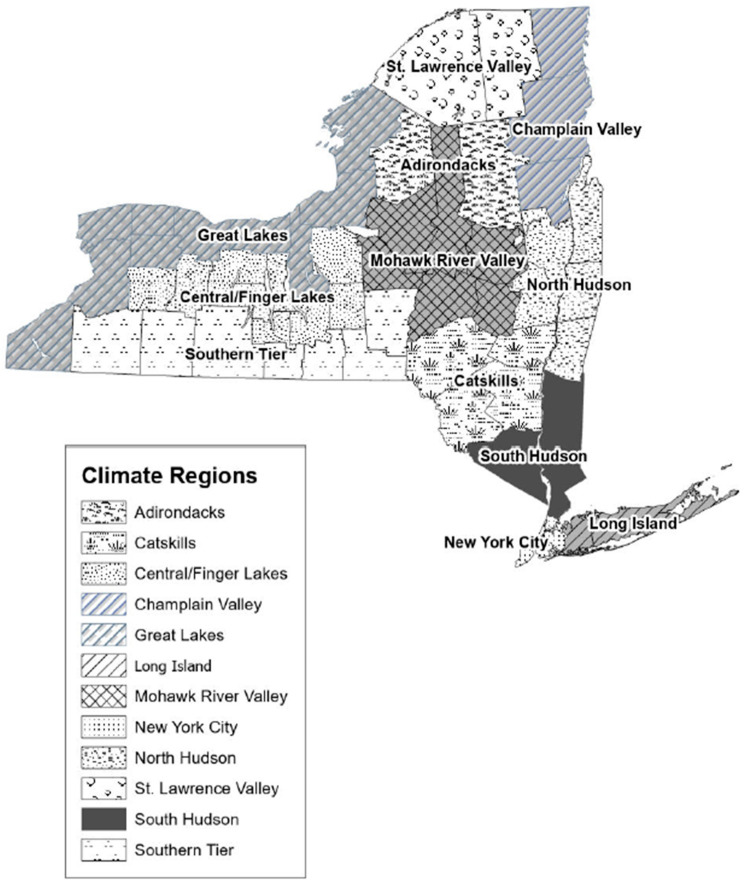
Map of NYS Climate Regions. Adirondacks: Lewis, Hamilton; Catskills: Delaware, Greene, Sullivan, Ulster; Central/Finger Lakes: Wyoming, Livingston, Ontario, Yates, Seneca, Schuyler, Tompkins, Cortland, Onondaga; Champlain Valley: Clinton, Essex, Warren; Great Lakes: Jefferson, Oswego, Cayuga, Wayne, Monroe, Orleans, Genesee, Niagara, Erie, Chautauqua; Long Island: Nassau, Suffolk; Mohawk River Valley: Madison, Oneida, Herkimer, Fulton, Montgomery, Otsego, Schoharie; New York City: Bronx, Queens, Richmond, New York, Kings; North Hudson: Columbia, Albany, Schenectady, Saratoga, Washington, Rensselaer; South Hudson: Orange, Putnam, Rockland, Westchester, Dutchess; Southern Tier: Cattaraugus, Allegany, Steuben, Chemung, Tioga, Broome, Chenango; St Lawrence Valley: St Lawrence, Franklin.

**Figure 2 ijerph-22-01124-f002:**
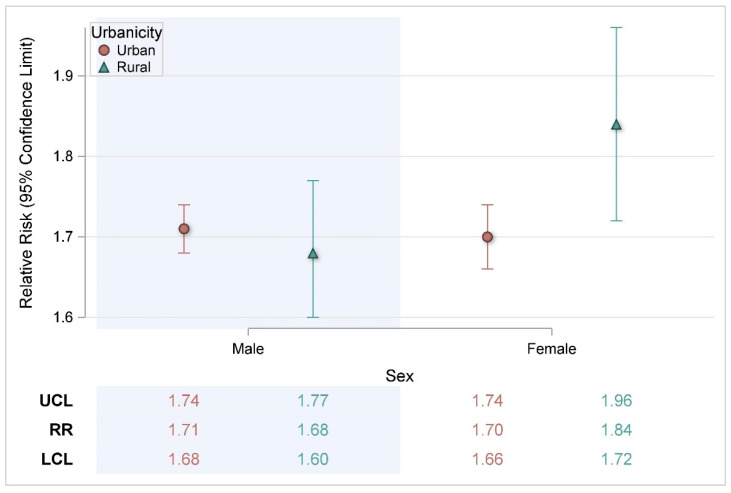
Risk of heat-related illness for a 5 °F change in heat index by urbanicity and sex. RR = Relative Risk; LCL = Lower Confidence Limit; UCL = Upper Confidence Limit.

**Figure 3 ijerph-22-01124-f003:**
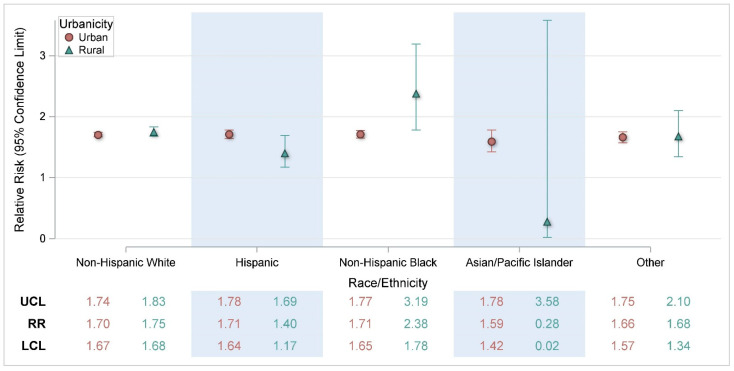
Risk of heat-related illness for a 5 °F change in heat index by urbanicity and race/ethnicity. RR = Relative Risk; LCL = Lower Confidence Limit; UCL = Upper Confidence Limit.

**Figure 4 ijerph-22-01124-f004:**
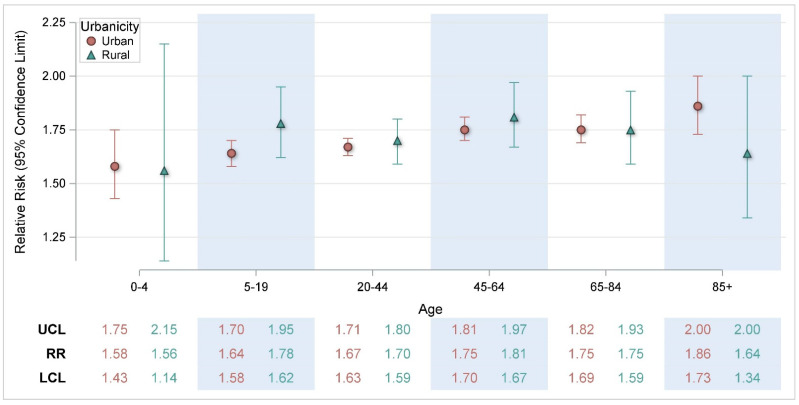
Risk of heat-related illness for a 5 °F change in heat index by urbanicity and age. RR = Relative Risk; LCL = Lower Confidence Limit; UCL = Upper Confidence Limit.

**Table 1 ijerph-22-01124-t001:** Distribution and rates of heat-related illness hospitalizations and ED visits in. New York State (May–September 2008–2019).

	Heat-Related Illness ^a^	
	*n* (%)	Rate ^b^
Cases	23,853 (23.49)	10.1
Control Days	77,678 (76.51)	
Inpatient	3247 (13.61)	1.4
Emergency Department	20,606 (86.39)	10.1
Case Month		
May	2186 (9.16)	0.9
June	5091 (21.34)	2.2
July	10,614 (44.50)	4.5
August	4085 (17.13)	1.7
September	1877 (7.87)	0.8
Age in years, Mean (SD)	43 (23.31)	
Age, years		
≤4	458 (1.92)	3.2
5–19	3911 (16.40)	9.0
20–44	8561 (35.89)	10.5
45–64	5964 (25.00)	9.4
65–84	3843 (16.11)	13.3
≥85	1116 (4.68)	22.2
Sex		
Male	13,992 (58.66)	12.2
Female	9861 (41.34)	8.1
Race/Ethnicity		
White, non-Hispanic	13,707 (57.46)	10.2
Hispanic	2889 (12.11)	6.7
Black, non-Hispanic	3994 (16.74)	11.8
Asian/Pacific Islander	380 (1.59)	2.0
Other ^c^	1577 (6.61)	26.9
Missing	1306 (5.47)	
Urbanicity ^d^		
Urban	20,685 (86.72)	10.0
Rural	3168 (13.28)	11.3
Climate Region ^e^		
Adirondacks	36 (0.15)	9.4
Catskills	552 (2.31)	13.0
Central/Finger Lakes	1391 (5.83)	12.7
Champlain Valley	337 (1.41)	15.1
Great Lakes	3509 (14.71)	11.6
Long Island	3819 (16.01)	11.1
Mohawk River Valley	899 (3.77)	13.2
New York City	7595 (31.84)	7.5
North Hudson	1386 (5.81)	11.9
South Hudson	2889 (12.11)	11.7
Southern Tier	1113 (4.67)	15.2
St. Lawrence Valley	324 (1.36)	16.5
Missing	3 (0.01)	

^a^ Heat-Related Illness: ICD-9 992, E900.0, E900.9; ICD-10 T67, X30, X32. ^b^ Annual Rate per 100,000 population using American Community Survey (ACS) 2011–2015. ^c^ Other Race: Native American, non-Hispanic multiracial, and non-Hispanic other races. ^d^ Rural–urban commuting area (RUCA) codes were used to categorize rural and urban areas of NYS. ^e^ Adirondacks: Lewis, Hamilton; Catskills: Delaware, Greene, Sullivan, Ulster; Central/Finger Lakes: Wyoming, Livingston, Ontario, Yates, Seneca, Schuyler, Tompkins, Cortland, Onondaga; Champlain Valley: Clinton, Essex, Warren; Great Lakes: Jefferson, Oswego, Cayuga, Wayne, Monroe, Orleans, Genesee, Niagara, Erie, Chautauqua; Long Island: Nassau, Suffolk; Mohawk River Valley: Madison, Oneida, Herkimer, Fulton, Montgomery, Otsego, Schoharie; New York City: Bronx, Queens, Richmond, New York, Kings; North Hudson: Columbia, Albany, Schenectady, Saratoga, Washington, Rensselaer; South Hudson: Orange, Putnam, Rockland, Westchester, Dutchess; Southern Tier: Cattaraugus, Allegany, Steuben, Chemung, Tioga, Broome, Chenango; St Lawrence Valley: St Lawrence, Franklin.

**Table 2 ijerph-22-01124-t002:** Risk of emergency visit/hospitalization for heat-related illness associated with a 5 °F change in maximum heat index in New York State (2008–2019).

	Heat-Related Illness ^a^
	RR ^b^ (CI ^c^)
All	1.71 (1.68, 1.73)
Month	
May	1.81 (1.72, 1.90)
June	1.69 (1.64, 1.74)
July	1.64 (1.60, 1.67)
August	1.86 (1.79, 1.94)
September	1.75 (1.66, 1.84)
Age	
0–4	1.59 (1.44, 1.75)
5–19	1.65 (1.60, 1.71)
20–44	1.67 (1.64, 1.71)
45–64	1.76 (1.71, 1.81)
65–84	1.75 (1.69, 1.81)
85+	1.83 (1.71, 1.96)
Sex	
Male	1.70 (1.67, 1.74)
Female	1.71 (1.68, 1.75)
Race/Ethnicity	
Non-Hispanic White	1.71 (1.68, 1.74)
Hispanic	1.69 (1.62, 1.76)
Non-Hispanic Black	1.72 (1.66, 1.78)
Asian/Pacific Islander	1.58 (1.41, 1.76)
Other ^d^	1.66 (1.58, 1.75)
Urbanicity ^e^	
Urban	1.70 (1.68, 1.73)
Rural	1.74 (1.67, 1.81)
Climate Region ^f^	
Adirondacks	3.28 (1.68, 6.42)
Catskills	1.69 (1.55, 1.84)
Central/Finger Lakes	1.75 (1.65, 1.85)
Champlain Vally	1.64 (1.46, 1.85)
Great Lakes	1.69 (1.63, 1.75)
Long Island	1.84 (1.76, 1.92)
Mohawk River Valley	1.71 (1.59, 1.84)
New York City	1.69 (1.65, 1.73)
North Hudson	1.63 (1.55, 1.72)
South Hudson	1.68 (1.61, 1.74)
Southern Tier	1.93 (1.80, 2.07)
St. Lawrence Valley	1.73 (1.54, 1.95)

^a^ Heat-Related Illness: ICD-9 992, E900.0, E900.9; ICD-10 T67, X30, X32. ^b^ Relative Risk; controlled for PM_2.5_ and ozone. ^c^ 95% confidence interval. ^d^ Other Race: Native American, non-Hispanic multiracial, and non-Hispanic other races. ^e^ Rural–urban commuting area (RUCA) codes were used to categorize rural and urban areas of NYS. ^f^ Adirondacks: Lewis, Hamilton; Catskills: Delaware, Greene, Sullivan, Ulster; Central/Finger Lakes: Wyoming, Livingston, Ontario, Yates, Seneca, Schuyler, Tompkins, Cortland, Onondaga; Champlain Valley: Clinton, Essex, Warren; Great Lakes: Jefferson, Oswego, Cayuga, Wayne, Monroe, Orleans, Genesee, Niagara, Erie, Chautauqua; Long Island: Nassau, Suffolk; Mohawk River Valley: Madison, Oneida, Herkimer, Fulton, Montgomery, Otsego, Schoharie; New York City: Bronx, Queens, Richmond, New York, Kings; North Hudson: Columbia, Albany, Schenectady, Saratoga, Washington, Rensselaer; South Hudson: Orange, Putnam, Rockland, Westchester, Dutchess; Southern Tier: Cattaraugus, Allegany, Steuben, Chemung, Tioga, Broome, Chenango; St Lawrence Valley: St Lawrence, Franklin.

## Data Availability

NLDAS heat index data are available from the New York State Environmental Public Health Tracking (EPHT) program epht@health.ny.gov (accessed on 29 June 2025). The health outcome data for this study came from the NYS Statewide Planning and Research Cooperative System (SPARCS) database. This data is not publicly available as it would compromise individual privacy. However, deidentified data is available from SPARCS upon request www.health.ny.gov/statistics/sparcs (accessed on 29 June 2025). PM_2.5_ and ozone data were obtained from the EPA DS model https://www.epa.gov/hesc/rsig-related-downloadable-data-files (accessed on 29 June 2025).

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
