# Peer review of "Assessing Heat–Health Vulnerability Through Temporal, Demographic, and Spatial Lenses: A Time-Stratified Case-Crossover Analysis in New York State"

_ijerph, 2025, doi:10.3390/ijerph22071124_

Round 1
Reviewer 1 Report
Comments and Suggestions for Authors
The manuscript investigates the combined effects of heatwaves, air pollution and green/ blue space on public health issues. This topic is significant for environmental science, climate change adaptation, and public health. While the study addresses an important public health issue, several areas require improvements to meet journal standards. I therefore recommend Major Revision.
Specific Comments:
1. Abstract
•The abstract is generally well-written and presents a clear and focused investigation for heart related health risks in NYS. However, a few sentences are overly dense and need to be simplified to enhance readability.
•The authors mention the potential intervention, and could outline specific recommendations or strategies that can support the implementation of related interventions.
2. Introduction
•Some sentences are overly long and need to be restructured to improve flow and readability.
•The rationale of examining the combinations of heatwaves with green/blue spaces is not clearly articulated. Please provide more explanation of the theoretical framework guiding this integrated approach.
3. Material and Methods
• Please provide a more detailed description of how the case-crossover design was implemented for this study. Explain its suitability for analyzing heat-related health risks, such as the reason why the ±7, ±14, and ±21-day control windows were selected and how this design improves the credibillty of the study.
• Please explain more about the temperature data acquisition, processing, and assignment procedure, and how this is related to individuals.
• The potential limitations need to be discussed, such as the study design, data sources, and analytical methods, and the authors should explain how these might influence the interpretation of the results.
4. Results
• When presenting comparisons, such as between rural and urban rates, the authors should state whether observed differences are statistically significant.
•The manuscript introduces risk ratios (such as RR = 1.71 overall), but these values need to be contextualized related to baseline risks.
5. Discussion
•This section reasonably interprets the results. More discussion about how the findings related to (agree, extend, or opposite) previous findings regarding air pollution's role in heat-health issues is needed.
•The discussion should acknowledge and evaluate potential limitations to strengthen the credibility of this study.
6. Conclusions
•This section should be more concise and synthesize key findings rather than repeating background information.
•The last section should also summarize this study’s key contributions, and provide suggestions for future research.
Author Response
|
Response to Reviewer 1 Comments
|
||
|
1. Summary |
|
|
|
Thank you very much for taking the time to review this manuscript. Please find the detailed responses below and the corresponding revisions in track changes in the re-submitted files.
|
||
|
2. Questions for General Evaluation |
Reviewer’s Evaluation |
|
|
Does the introduction provide sufficient background and include all relevant references? |
Yes |
|
|
Is the research design appropriate? |
Can be improved |
|
|
Are the methods adequately described? |
Can be improved |
|
|
Are the results clearly presented? |
Yes |
|
|
Are the conclusions supported by the results? |
Can be improved
|
|
|
Are all figures and tables clear and well-presented? |
Yes |
|
|
3. Point-by-point response to Comments and Suggestions for Authors |
||
|
Comments 1: Abstract |
||
|
Response 1: Thank you for pointing this out. We have simplified some of the sentences and included specified recommendations, as suggested. Given the limited word count in the abstract, more detail is provided in the discussion. See Line 12-23.
|
||
|
Comments 2: Introduction Response 2: We have edited the longer sentences for clarity. We did not assess green/blue spaces in this analysis specifically. However, urban/rural risks and climate region risks were explored. Please see added sentences for our rationale for inclusion. Line 27-60.
|
||
|
Comments 3: Methods • Please provide a more detailed description of how the case-crossover design was implemented for this study. Explain its suitability for analyzing heat-related health risks, such as the reason why the ±7, ±14, and ±21-day control windows were selected and how this design improves the credibility of the study. Response 3a: Thank you for your review and commentary, we may need additional clarification on the detailed description for the case-crossover analysis beyond what is outlined in the Statistical Analysis section. Line 141-167. The reasoning for “±7, ±14, and ±21-day control windows” are detailed in the following sentence “This control period selection identifies control days restricted to the same day of the week, month, and year to control for seasonal and day of week variation” Line 148-150. We have added language on how the case-crossover study design is ideal for heat-health analysis. Line 151-154.
Response 3b: Please refer to the Air Temperature section of the Methods where we discuss in detail data acquisition, processing, and assignment to individual data by 2010 Census tracts. If there is any detail that you feel is important for us to include beyond what is described, please let us know. Line 83-101.
Response 3c: A potential limitation of case-crossover analysis is confounding from time varying factors. We added detail on this in addition to how time stratification (““±7, ±14, and ±21-day control windows”) and including potential confounders in the model addresses this. Line 155-157. Other limitations, such as data sources are presented in the discussion. Line 398-414.
|
||
|
Comments 4: 4. Results Response 4a: We agree. Due to the high population counts in the denominators for the rates, confidence intervals are very narrow. Therefore, rates are statistically significantly different. We have added this to the manuscript at Line 181-182.
Response 4b: In the case-crossover study design the baseline risk is not estimated in the way it would be in a cohort or case-control study. The case-crossover study design compares exposure during case period to exposure during the control periods when the event did not occur (within the same individual). The baseline risk is assumed to be constant within the individual, given the short time window around the hazard and control period. In the case of this study, within a maximum of ±21 days. Even without baseline risk, the RR provides valuable information about the magnitude of the exposure's effect, which can guide public health interventions.
|
||
|
Comments 5: Discussion Response 5a: Our study primarily focuses on heat-related illness rather than diseases such as asthma or cardiovascular conditions, which are more directly linked to air pollution exposure. The inclusion of air quality variables (e.g., PM2.5 and ozone) in our models was intended to control for potential confounding rather than to explore the independent or synergistic effects of air pollution on health outcomes. This approach ensures that the observed associations between heat exposure and heat-related illness are not biased by the potential influence of air pollution. Response 5b: There is a limitations paragraph that discusses data sources, stratification and small counts, and exposure assignment issues. We have added some additional limitations considerations. Line 398-414.
|
||
|
Comments 6: Conclusions •The last section should also summarize this study’s key contributions, and provide suggestions for future research. |
||
|
Response 6: We edited the conclusion to be more concise. We included a synthesis of findings, their relevance, and future efforts. Line 420-439.
|
||

Reviewer 2 Report
Comments and Suggestions for Authors
Thank you for a well-written paper. I have a few small suggestions that I think will improve the paper.
Abstract: Delete semicolon after age in line 12
Introduction:
Line 33: add also - may also experience profound
Line 44: Delete comma after such as
Line 52: (NWS) to lower the heat advisory threshold (delete the prior "heat advisory")
Line 56: Delete the sentence that starts: "We aim to assess"
Materials and Methods:
Why not 2020 Census tracts?
Line 112 - 125: Why are some of these bold?
Statistical Analysis: Line 144: Change week day to "day of week"
Results:
Line 189: Throughout there is no need to keep saying "significant." Either say statistically significant or just delete this word. Along the same lines, there is no need to report the p-values when the CIs are given. They convey the same information with the CIs offering more information in fact.
Line 196: Please change numbers to counts.
Discussion:
Line 255: within the state. These findings may assist with facilitating (Add a period after state and reword the start of the next sentence)
Line 265-272: It should also be mentioned that many may view May as too early in the summer to use air conditioning even if it is available. This citation may be appropriate to review and include: Jian, Y., Liu, J., Pei, Z., & Chen, J. (2022). Occupants’ tolerance of thermal discomfort before turning on air conditioning in summer and the effects of age and gender. Journal of Building Engineering, 50, 104099.
Line 298: Please start a new paragraph with "Conversely, in rural areas..."
Line 315: please delete paucity and change to little published research examining the health...
Line 323: Please add how this may affect the results, such as by adding something about wider confidence intervals or less accuracy
Line 352-358: I suggest that this short paragraph is deleted. It doesn't add anything to the discussion.
Line 358: Change leverage to use
Conclusion: Line 381: Is important for targeted public health interventions
Line 389: please change targeted at to targeted to
Line 389: Please reword this sentence: Public health messaging and interventions such as the HeatRisk tool developed by the NWS in partnership with the CDC can be useful early-warning resources.
Author Response
|
Response to Reviewer 2 Comments
|
||
|
1. Summary |
|
|
|
Thank you very much for taking the time to review this manuscript. Please find the detailed responses below and the corresponding revisions in track changes in the re-submitted files.
|
||
|
|
|
|
2. Questions for General Evaluation |
Reviewer’s Evaluation |
|
Does the introduction provide sufficient background and include all relevant references? |
Yes |
|
Is the research design appropriate? |
Yes |
|
Are the methods adequately described? |
Yes |
|
Are the results clearly presented? |
Yes |
|
Are the conclusions supported by the results? |
Yes
|
|
Are all figures and tables clear and well-presented? |
Yes |
- Point-by-point response to Comments and Suggestions for Authors
Abstract Comments:
Delete semicolon after age in line 12
Response: Accepted edit. Thank you for pointing this out.
Introduction Comments:
Line 33: add also - may also experience profound
Line 44: Delete comma after such as
Line 52: (NWS) to lower the heat advisory threshold (delete the prior "heat advisory")
Line 56: Delete the sentence that starts: "We aim to assess"
Response: Accepted all edits.
Materials and Methods Comments:
Why not 2020 Census tracts?
Response: We used 2010 Census tracts because the study period (2008 - 2019) was prior to the adoption of 2020 Census tracts. Additionally, 2010 tracts were useful in maintaining consistency for linking datasets since the NLDAS air temperature dataset, EPA downscaler air pollution data, and the ACS 2015 data all use 2010 tracts.
Line 112 - 125: Why are some of these bold?
Response: The bold text indicates the climate regions, and the regular text are the counties within the regions.
Statistical Analysis: Line 144: Change week day to "day of week"
Response: Accepted edit.
Results Comments:
Line 189: Throughout there is no need to keep saying "significant." Either say statistically significant or just delete this word. Along the same lines, there is no need to report the p-values when the CIs are given. They convey the same information with the CIs offering more information in fact.
Response: Agree. We added “statistically” throughout. See Line 182-277. The p-values were removed and figures were updated.
Line 196: Please change numbers to counts.
Response: Accepted edit.
Discussion Comments:
Line 255: within the state. These findings may assist with facilitating (Add a period after state and reword the start of the next sentence)
Line 265-272: It should also be mentioned that many may view May as too early in the summer to use air conditioning even if it is available. This citation may be appropriate to review and include: Jian, Y., Liu, J., Pei, Z., & Chen, J. (2022). Occupants’ tolerance of thermal discomfort before turning on air conditioning in summer and the effects of age and gender. Journal of Building Engineering, 50, 104099.
Response: Added reference to early heat season heat acclimation behavior with citation. See Line 296-298.
Line 298: Please start a new paragraph with "Conversely, in rural areas..."
Line 315: please delete paucity and change to little published research examining the health...
Line 323: Please add how this may affect the results, such as by adding something about wider confidence intervals or less accuracy
Line 352-358: I suggest that this short paragraph is deleted. It doesn't add anything to the discussion.
Response: We agree this does not add to the discussion as it is more summative. We integrated the paragraph into the conclusion. See Line 431-436.
Line 358: Change leverage to use
Response: Accepted all other comments and edits.
Conclusion Comments:
Line 381: Is important for targeted public health interventions
Line 389: please change targeted at to targeted to
Line 389: Please reword this sentence: Public health messaging and interventions such as the HeatRisk tool developed by the NWS in partnership with the CDC can be useful early-warning resources.
Response: Moved to Line 383-385 where it fits better with discussion of adaptation efforts.
Response: A

Round 2
Reviewer 1 Report
Comments and Suggestions for Authors
I have carefully reviewed the revised manuscript and the author's response to the previous comments. The revisions have significantly improved the clarity, coherence, and overall quality of the paper. Given the significant improvement, I recommend that the manuscript be accept in current form.